# The Effects of Meditation with Stabilization Exercise in Marine Region on Pain, Tactile Sense, Muscle Characteristics and Strength, Balance, Quality of Life, and Depression in Female Family Caregivers of People with Severe Physical Disabilities: A Randomized Single-Blinded Controlled Pilot Study

**DOI:** 10.3390/ijerph19106025

**Published:** 2022-05-16

**Authors:** Ho-Jin Shin, Sung-Hyeon Kim, Hwi-Young Cho, Jae-Hon Lee

**Affiliations:** 1Department of Health Science, Gachon University Graduate School, Incheon 21936, Korea; sports0911@hanmail.net (H.-J.S.); q315201@naver.com (S.-H.K.); 2Department of Physical Therapy, Gachon University, Incheon 21936, Korea; 3Department of Psychiatry, Schulich School of Medicine and Dentistry, Western University, London, ON N6A 5W9, Canada

**Keywords:** marine therapy, meditation, stabilization exercise, caregivers, mental health, physical health

## Abstract

Female caregivers of people with disabilities are burdened physically and mentally. To improve these symptoms, an intervention that is easy to apply and has fewer side effects, such as natural healing, has been proposed, but the effect of healing using marine resources is unclear until now. The purpose of this study is to investigate the effect of meditation accompanied with stabilization exercise in the marine region on the improvement of pain, tactile sense, muscle characteristics, muscle strength, balance, quality of life, and depression in female caregivers of people with severe physical disabilities. Twenty-four female family caregivers were recruited and were randomly assigned to the marine therapy group (MTG, *n* = 12) and the control group (CG, *n* = 12). Both groups performed the same meditation (35 min) and stabilization exercise (25 min) twice a day for 3 nights and 4 days per session (total 8 sessions). The MTG performed these in the marine region, whereas the CG performed the interventions in the urban region. Pain (pain intensity and pain pressure threshold), tactile sense (tactile spatial acuity), muscle characteristics (stiffness, elasticity), muscle strength (hand and pinch grip strength), balance, quality of life, and depression were measured before and after the intervention and 4 weeks after the intervention. Both groups showed significant improvements in pain intensity (resting pain: f(2) = 72.719, *p* < 0.001; movement pain: f(2) = 24.952, *p* < 0.001), muscle strength (right pinch grip: f(2) = 15.265, *p* < 0.001), and depression (f(2) = 13.312, *p* < 0.001), while tactile spatial acuity (TSA) (upper part: f(2) = 14.460, *p* < 0.001; lower part: f(2) = 7.672, *p* = 0.002), dynamic balance (f(2) = 4.196, *p* = 0.024), and quality of life (overall quality of life & general health: f(2) = 5.443, *p* = 0.009; physical health: f(2) = 13.991, *p* < 0.001; psychological: f(2) = 9.946, *p* < 0.001; environmental: f(2) = 20.004, *p* < 0.001; total: f(2) = 11.958, *p* < 0.001) were significantly improved only in MTG. There was no significant change in pain pressure threshold (upper trapezius (UT): f(2) = 0.765, *p* = 0.473; levator scapula (LS): f(2) = 0.213, *p* = 0.809; splenius capitis (SC): f(2) = 0.186, *p* = 0.831) and muscle characteristics (UT stiffness: f(2) = 1.486, *p* = 0.241; UT elasticity: f(2) = 0.358, *p* = 0.702; LS stiffness: f(2) = 2.440, *p* = 0.102; LS elasticity: f(2) = 0.544, *p* = 0.585) in both groups. In comparison between groups, the MTG showed a significant difference in sensory function compared to the CG (resting pain: f(2) = 10.487, *p* = 0.005; lower part: f(2) = 5.341, *p* = 0.034 in TSA). Our findings suggest that meditation combined with stabilization exercise improved pain, muscle strength, and depression of female caregivers. In particular, greater benefits on tactile sense, balance, and quality of life were found in performing these in the marine region compared to the urban region.

## 1. Introduction

The number of long-term people with disability for neurological diseases such as stroke and spinal cord injury is constantly increasing worldwide [1]. Most of the care for family members with disabilities is informal, and these care and medical services in hospitals and nursing homes are essential for the long-term care of people with disability [2,3]. 

The family caregiver who takes care of people with long-term physical disability experiences mental stress due to lack of sleep, insufficient exercise, and absence of rest, and also has a physical burden (fatigue, backache, physical strain) due to the frequent disabled people transfer and uncomfortable postures of the caregivers [4,5,6,7]. Musculoskeletal pain and psychological stress lead to an increase in physical or mental health problems and a decrease in the quality of life (QoL) of the caregiver. In addition, the depressive emotion of people with disability can become a factor that decreases the QoL of the caregiver [8]. This has negative consequences for the care of people with disability and their families, and it is necessary to identify interventions that can improve and restore the physical and mental health of caregivers. 

The social roles of women and men have also changed along with the changes in society and industry. Still, in Korea, the gender-specific norms continue in which men are responsible for economic support and women take on various household chores. Therefore, the role of caring for a family with physical or mental disabilities is mainly taken by women, and many women experience physical or mental burdens due to long-term care [9,10]. Unfortunately, however, most studies to date have focused on treatment or symptom improvement methods for people with disabilities, and studies on the health promotion of their family caregivers are very scarce.

Therapeutic interventions such as meditation, yoga, and physical exercise can regulate the sympathetic nervous system and hypothalamic–pituitary–adrenal system [11] and effectively relieve work-related stress and associated illness and burnout [12]. These have also been used to improve physical problems and function, pain, and fatigue; meditation is commonly used to reduce mental stress and depression in clinics and health care facilities. Some studies have reported that these interventions positively affect managing QoL, stress, depression, and physical burden for caregivers [13,14,15]. According to Kaplan’s attention restoration theory, exposure to the forest or wild nature is effective for mental health. This nature could improve the ability of urban dwellers, stress, and deteriorated performance by providing a stress-free environment and a landscape of a pattern where they can concentrate without much attention [16]. These results indicate that the environment can be an important factor in the subject’s sensory function and psychological changes. Recently, natural therapy has been proposed to provide safe and alternative treatment and reduce side effects or boost the effectiveness of traditional medicine. In particular, marine resources such as seawater, marine sand, sea breezes, and marine scenery positively affect mental and physical health [17,18,19]. In particular, the marine area is a rich environment with various kinds of resources such as sea sands, ocean sound, marine climate, seawater, and sea scenery. It has been reported that natural ocean sound exposure and ocean-side relaxation improved hearing difficulty and stress hormone levels for chronic tinnitus patients [17]. Climatotherapy at the Dead Sea also positively affected QoL, depression, and stress in the elderly and patient groups with poor physical health, QoL, and depression [20]. Our previous study confirmed that therapeutic exercise on sea sand positively improved pain and motor function in people with chronic ankle instability. This is considered to be an improvement through various sensory stimulation of the feet and activation of muscle and joint proprioceptive sensations by sand. In addition, the unstable surface created due to the mechanical properties of sand requires more neuromuscular recruitment and control for the subject to perform motions. As such, sea sand requires an increased energy cost to perform a movement compared to grass or flat land, thereby improving cardiopulmonary function. It has the advantage that muscle damage, pain, and performance can be improved due to a relatively low impact given to the joint [19,21]. Although it has been reported that marine healing effectively improves musculoskeletal and psychological problems, the effect of marine healing on family caregivers who suffer from these problems has not been identified.

The two objectives of this study are as follows. First, it investigates the effect of meditation with stabilization exercise on pain, tactile sense, muscle characteristics and strength, balance, QoL, and depression in female family caregivers of people with severe physical disabilities. Second, it is also to investigate whether the effect of meditation with stabilization exercise performed in the marine region is more effective than the intervention performed in the urban region.

## 2. Materials and Methods

### 2.1. Participants

We recruited family caregivers who care for people with severe physical disabilities at the National Rehabilitation Center. We posted information about the research introduction and participants’ recruitment on the hospital’s bulletin board and the online community of caregivers. Forty female family caregivers volunteered to participate in this study, and 24 subjects finally participated in consideration of the inclusion and exclusion criteria. The participants who participated in this study were randomly assigned to the marine therapy group (MTG) or the control group (CG). The inclusion criteria were as follows: (1) aged >19 and <65, a female caregiver who cares for people with severe physical disability (e.g., stroke (over moderate ≥9 NIHSS (National Institutes of Health Stroke Scale), or traumatic brain injury) [22,23], (2) non-specific low back pain and neck pain in which a neurologist detected no neuropathological problem or structural damage, and (3) who has confirmed subjective stress due to caring through an interview with one psychiatrist (HDRS score ≥8) [22]. The excluded from the study were female caregivers with the following characteristics: (1) with neurological, psychiatric, visual, auditory, cardiovascular, and musculoskeletal disorders to the degree that makes it difficult to carry out relaxation exercises; (2) those who are accompanied by specific back pain that appears due to causes such as nerve root pain, infection, spinal cord injury, trauma, cauda equina syndrome, and structural deformation; (3) those who complained of shoulder pain caused by accident within four weeks before the experiment; (4) those who complained of pain due to acute inflammation; (5) those in whom pain manifests through the neck extension movement; and (6) those who have experienced upper limb fracture, dislocation or surgery within six months.

To calculate the sample size, we used the G*power program (G*power ver. 3.1.9.2, University of Kiel, Aichach, Germany). No previous study has considered this study’s intervention methods and outcome measurements, so we performed a preliminary study to estimate the sample size. Therefore, the sample size was calculated by selecting ANOVA: Repeat Measures, within-between interaction of F Test, and entered an effect size f = 0.26, α error Prob = 0.05, Power = 0.8, Number of Groups = 2, Number of Measurements = 3 into the program. A total of 26 persons were calculated, 13 in each group. A total of 29 persons were determined by applying a 10% dropout rate for the duration of the study.

Of the 29 subjects who satisfied the selection criteria of this study, five were dropped, and a total of 24 subjects finally participated in the study. In the MTG, three people dropped out during the study period due to personal reasons, and in the CG, two subjects dropped out due to health problems (Figure 1). According to the parameters, there were no significant inter-group differences in age (*p* = 0.774), height (*p* = 0.669), weight (*p* = 0.364), BMI (*p* = 0.245), care period (*p* = 0.396), and relationship (*p* = 0.974) of the participant characteristics (Table 1). 

Before the study, the participant was explained about this study and signed a consent form. This study was conducted with the approval of the Gachon University Institutional Review Board (1044396-201907-HR-129-01). This clinical trial was registered with the Clinical Research Information Service (CRIS, the Korean Registry of Clinical Trials) (CRIS registration number: KCT0005335).

### 2.2. Procedure

This study was designed as a randomized single-blinded controlled pilot study. The MTG stayed at the Dukgu Hot Spring Resort for three nights and four days and performed meditation and exercise on the Bugu Beach and Hujeong Beach in September 2019 (Uljin-gun, Gyeongsangbuk-do, marine region). On the other hand, the CG performed the same intervention in the National Rehabilitation Center for four days and three nights and stayed at a nearby hotel in October 2019 (Seoul, urban region). Both groups were trained to engage in the same amount of physical activity during and outside work hours. In order not to burden the subject with family care, visits from the participant’s family were not allowed.

All participants were allowed one and half hours of light walks around the hotel during off-hours. The MTG group performed this and viewed marine scenery on the beach or the road around the shore, and the CG group took a walk or appreciated the city park scenery around the hotel.

We performed a total of theee measurements before applying the first intervention (pre-intervention, PRE), after the last intervention (post-intervention, PI), and four weeks after the last intervention (follow-up, FU). PI measurement was conducted the next day after the intervention was completed, considering a single-trial effect.

### 2.3. Measurement

The primary outcomes were QoL, depression, pain (pain intensity, and pain pressure threshold (PPT)), and the secondary outcomes were tactile sense (tactile spatial acuity, TSA), muscle characteristics, muscle strength (hand and pinch grip strength), and balance (dynamic balance).

#### 2.3.1. Pain Intensity

Pain intensity was measured using a 100 mm-visual analog scale [24]. After explaining the measurement tool sufficiently to each participant, the assessor asked the participant to check the pain that was felt from 0 (no pain) to 100 (worst possible pain). The evaluation was made after presenting two situations (resting pain and movement pain). Resting pain usually refers to pain when resting without moving the body, and movement pain refer to pain that occurs during daily life, such as gait and housework. A more than 20 mm reduction is considered the minimal clinically important difference [25].

#### 2.3.2. Pain Pressure Threshold

A digital algometer (Somedic AB, Farsta, Sweden) with a 1 cm^2^ probe was used to measure PPT [26]. The measurement was performed after marking the application position of the targeted muscle with a pen. The probe was applied vertically to the skin above the selected muscle at a constant speed of 30 kPa/s, and the participant was instructed to express the feeling of pain (grimace, sound, avoidance, increased muscle tone) as soon as the pressure sensation was felt like pain during the measurement. The measurement was repeated three times, and the interval between measurements was set to 30 s. The selected muscles were the upper trapezius (UT), levator scapulae (LS), splenius capitis (SC), and erector spinae muscle (ES), and the application location was set as the myofascial trigger point of each muscle [27].

#### 2.3.3. Tactile Spatial Acuity

TSA was measured with an esthesiometer (Two-Point Discriminator, Baseline^®^, Fabrication Enterprises, White Plains, NY, USA) [28]. TSA was measured in a horizontal direction in the cervical region and the upper trapezius. With the participant lying down with their eyes closed, the assessor applied stimulation five times to decrease the distance between the cervical region and the upper trapezius region with two points or one point of the esthesiometer. The participant answered whether they recognized one or two points and one point if they were uncertain during the measurement. The first interval started at 4.5 cm (cervical region) and 7 cm (upper trapezius) and was decreased by 0.5 cm until the participant detected only one point. The final distance was used as data by increasing the distance by 0.2 cm based on the point detected by one point. To prevent speculation due to the pattern, stimulation was carried out with one or two chosen points at random. While each stimulus created an indentation in the skin, none of the participants complained of any discomfort.

#### 2.3.4. Muscle Characteristics

Muscle characteristics were measured by using a handheld myotonometer (Myoton AS, Tallinn, Estonia) [29,30]. The measurement region was marked with a marker before measurement, and the measurement was repeated. The participant was prone and the assessor applied the probe perpendicular to the skin over the selected muscle. The values measured five times were used as data. The stiffness (N/m) and elasticity (logarithmic decrement, Log D) of the four muscles (UT, SC, LS, paraspinal muscle of L2, L3) were measured.

#### 2.3.5. Muscle Strength

A digital dynamometer (Jamar Plus; Sammons Preston, Rolyon, Bolingbrook, IL, USA) and a pinch gauge (Jamar pinch gauge; Patterson Medical, Bolingbrook, IL, USA) were used to measure muscle strength [31,32]. Grip strength was measured by using a digital dynamometer, and the pinch gauge was used to measure pinch grip strength. To measure grip strength, the participant was instructed to maintain a state of shoulder adduction to 90°, the elbow flexion to 90°, and the wrist slight extension (0–30°) while the upright sitting position on the back of a chair. The participant was asked to grip with maximum force for three seconds while holding the measuring equipment. If excessive abduction of the shoulder, distortion of the trunk, and deviation of the neutral position of the wrist were found, it was considered a measurement error and remeasurement was performed. The measurement was repeated three times, and the interval between measurements was 1 min. Pinch grip strength measurement was performed in the same method as grip strength measurement.

#### 2.3.6. Dynamic Balance

Time up and go (TUG) was performed to measure dynamic balance. The measurement began with the participant in a sitting position. With verbal instruction given by the assessor, the participant had to get up from the chair, walk 3 m as quickly and safely as possible, go around a pillar, return to the chair, and sit down again. The measurement began at the time when the participants stood up from the sitting position and ended when the participant’s buttocks touched the chair after sitting down again. The measurement was repeated three times using a timer, and the interval between measurements was set to 30 s [33].

#### 2.3.7. Quality of Life

QoL is defined as an individual’s perception of their position in life in the context of the culture and value systems in which they live and concerning their goals, expectations, and concerns. Although QoL is a subjective concept in nature, it is being used as a standard health outcome measure by evaluating factors related to people’s mental, physical and social health. We used the Korean version of the World Health Organization quality of life instrument-short version (K-WHOQOL-BREF), with a Cronbach’s alpha of 0.898 [34,35]. Each item of the measurement tool is scored on a Likert scale (1 = disagree, 5 = completely agree), and higher scores indicate a more positive quality of life. The measurement tool is composed of 26 questions with two questions for the Overall QOL & General Health domain, seven questions for the Physical Health domain, six questions for the Psychological domain, three questions for the Social Relationships domain, and eight questions for the Environmental Domain. The score for each domain was converted to a perfect score of 100 [34]. Questions 3, 4, and 26 that are related to being positive were inverted and summed with the other questions to be scored. The total score was converted into 100 points for each domain. A higher score means that the perception or satisfaction of the individual for the corresponding domain is increased. The Cronbach’s alpha measured in this study was 0.94.

#### 2.3.8. Depression

Depression is a mood disorder characterized by persistent sadness or loss of interest, which is considered a common mental disorder in caregivers. It was measured using the Korean version-Hamilton Depression Rating Scale (K-HDRS), with Cronbach’s alpha of 0.76 [36,37]. This measurement tool is composed of 17 questions. 9 items (depressed mood, feeling of guilt, suicide, work and activities, retardation, agitation, psychic anxiety, somatic anxiety, hypochondriasis) scored as 0–4 points, and 7 items (insomnia early, insomnia middle, insomnia late, somatic gastrointestinal, somatic general, genital symptoms, insight) were scored as 0–2 points, and one item (loss of weight) was scored 0–3 points. The total score is 53 points, and a higher score means that the depression is more serious. The Cronbach’s alpha measured in this study was 0.77.

### 2.4. Intervention

The intervention applied to this study was performed a total of 8 times, with two sessions per day for four days and three nights and 60 min per session. The intervention consisted of meditation and a stabilization exercise to promote physical activity.

#### 2.4.1. Meditation (35 min)

The meditation was reorganized by referencing the Menezes protocol [38]. The composition of the meditation performed in this study is as follows: (1) diaphragm breathing exercise (5 min), (2) meditation (20 min), (3) closing with discussion and exchange of thinking (10 min). 

For the warm-up exercise, the subjects stood in a circle facing each other and performed diaphragm breathing with light stretching. Meditation was performed in a sitting position while maintaining the circle to meet each other. After the meditation, each subject had time to discuss their feelings and thoughts during the day. The rest time between interventions was 10 min, and one physical therapist and one instructor who had acquired a meditation certificate conducted the intervention.

#### 2.4.2. Stabilization Exercise (25 min)

The stabilization exercise that was applied in this study was composed of a core exercise that improves the stability of the trunk and a disassociation exercise that can train the movement of the limbs while maintaining a neutral spine position. They can reduce the musculoskeletal burden and improve faulty movement patterns in daily life. The stabilization exercise that was performed in this study was reorganized based on previous research [39]. The stabilization exercise was performed after a light warm-up exercise. This exercise consisted of 5 exercises (plank, bridge, bird dog, dead bug, side plank) performed for 5 min for each exercise.

### 2.5. Statistics Analysis

The data was analyzed using the IBM SPSS Statistics 25.0. (IBM-SPSS Inc, Chicago, IL, USA). Descriptive statistics for demographic data and all outcome measures were expressed as averages and standard deviations. A test of normality was performed using the Shapiro–Wilk test. As a result, parametric statistics were performed since they followed a normal distribution. For the homogeneity test, continuous variables were analyzed with the independent *t*-test, and categorical variables were analyzed with the chi-square test. Repeated-measures ANOVA was used to analyze the changes in variables between the groups (MTG, CG) over time (PRE, PI, FU). The independent *t*-test was used to compare two groups, and the paired *t*-test was used to confirm the difference over time (PRE, PI, FU) within the group. The Bonferroni correction was applied to prevent Type 1 error. The alpha level was set to 0.05.

The effect size was estimated with Cohen’s d and f test. Cohen’s d effect size was expressed as small ≥0.2, medium ≥0.5, large ≥0.8, and the partial eta squared (η^2^) effect size was expressed as small ≥0.10, medium ≥0.25, large ≥0.40 [40].

## 3. Results

### 3.1. Pain Intensity

Compared to PRE, in the MTG, resting pain and movement pain both showed a significant decrease at PI and FU (*p* < 0.05), whereas in the CG, resting pain showed a significant decrease only at PI (*p* < 0.05). The MTG also showed significant improvements at PI and FU in resting pain and movement pain than the CG (*p* < 0.05). Statistically significant time × group interaction was confirmed in resting pain and movement pain between groups (*p* < 0.05) (Table 2).

### 3.2. Pain Pressure Threshold

There was no significant difference before and after intervention in UT, LS, and SC of the MTG and CG, and there was no significant difference in comparison between the two groups (*p* > 0.05) (Table 2).

### 3.3. Tactile Spatial Acuity

Compared to PRE, the MTG showed a significant improvement in the upper part and lower part at PI and FU (*p* < 0.05), whereas the CG showed no significant change after the intervention (*p* > 0.05). In the comparison between the two groups, the upper part showed a significant difference only at FU (*p* < 0.05), but the lower part showed a significant difference at PI and FU (*p* = 0.004). There was a significant time × group interaction in the lower part between the two groups (*p* < 0.05) (Table 3).

### 3.4. Muscle Chatracteristics

After the intervention in both groups, there was no significant change in stiffness and elasticity of the three muscles, and there was no significant difference between the two groups at the three measurement time points (*p* > 0.05) (Table 4).

### 3.5. Muscle Strength

The MTG showed a significant improvement in the grip strength of both hands at PI and FU time (*p* < 0.05), whereas the CG group showed no significant change after the intervention (*p* > 0.05). In the pinch grip, the MTG showed a significant increase in the right and left hands at PI (*p* < 0.05), but there was no significant change at FU (*p* > 0.05). On the other hand, the CG showed a significant difference only on the right side after intervention (*p* < 0.05), and there was no significant change in this hand at FU and the left side at PI and FU (*p* > 0.05). There was no significant difference in muscle strength between the two groups at any time (*p* > 0.05) (Table 5).

### 3.6. Dynamic Balance

In TUG, both groups showed no significant changes before and after the intervention, and there was no significant difference between groups at the three measurement points (*p* > 0.05) (Table 5).

### 3.7. Quality of Life

The MTG showed significant improvements over time in all variables except social relationships among WHOQOL (*p* < 0.05), whereas the CG showed no significant change in all variables (*p* > 0.05). There was also no significant difference between the two groups at all measurement time points (*p* > 0.05). WHOQOL-physical health, WHOQOL-environmental, and WHOQOL-total showed significant time × group interaction between the two groups (*p* < 0.05) (Table 6).

### 3.8. Depression

Pre-intervention, there were no significant differences between the groups (*p* = 0.987). There was no significant time × group interaction in all variables (*p* > 0.05). In the MTG, there was a significant difference over time (*p* < 0.05). There was a significant difference between pre vs. PI (*p* = 0.018). In the CG, there was a significant difference over time (*p* < 0.05). There a was significant difference between pre vs. PI (*p* = 0.042). In a comparison between groups, there were no significant differences in all variables (*p* > 0.05) (Table 6).

## 4. Discussion

This study investigates the effects of the marine healing program using meditation combined with stabilization exercise on pain, tactile sense, muscle characteristics, muscle strength, balance, quality of life, and depression in female caregivers of people with severe physical disabilities.

Both groups significantly improved resting pain, right pinch grip strength, and depression, and showed a large (resting pain: η^2^ = 0.81, depression: η^2^ = 0.44) and medium (right pinch grip strength: η^2^ = 0.24) effect size. In particular, the MTG showed significant improvement in movement pain, tactile sense, bilateral hand & pinch grip strength, dynamic balance, and QoL. In addition, the effect of intervention lasted for a month in all variables except right pinch grip strength. The previous study suggested the positive effects of meditation and exercise in improving college students’ emotions and attention regulation [38]. We modified these methods by considering the characteristics of female caregivers. 

Although the two groups performed the same interventions, the effect size in resting pain (MTG: *d* = 2.65, CG: *d* = 1.50) and movement pain (MTG: *d* = 2.08, CG: *d* = 0.51) was different, and the follow-up effect was also different in resting pain (MTG: *d* = 2.27, CG: *d* = 0.45) and movement pain (MTG: *d* = 1.75, CG: *d* = 0.30). Stabilization exercise reduces pain by improving the function of the trunk and the respiratory function [39], enhances the quality of distal limb movement [41,42], and can help induce maximum muscular strength [43]. It has also been reported to effectively alleviate non-specific neck pain and shoulder pain [44,45,46,47,48] and improve physical functions such as muscle strength, balance ability, and range of motion [49,50,51,52]. Meditation can also reduce pain in patients with chronic pain [53]. It effectively reduces pain in various target groups, such as low back pain, fibromyalgia, and musculoskeletal pain [54,55,56]. Meditation causes mental and physical stability, inhibition of the sympathetic nervous system activity, and decreased automatic parameters such as heart rate, respiration rate, and adrenergic responsiveness [57,58]. Sympathetic nerve activity can increase pain in the pathological state [59], and it is believed that a decrease in sympathetic nerve activity through meditation reduces pain. 

A previous study [60] showed that chronic low back pain was reduced by 38.55% by applying stabilization exercise to caregivers five times a week for eight weeks. In the pain of our results, the CG showed no significant improvements, whereas the MTG showed 66.5–71.0% pain relief despite short-term intervention. Through this, it can be inferred that the environmental factor, marine scenery, can be an important mediating variable in improving pain. In addition, compared to previous studies, the CG of this study showed no improvement in all variables except pain intensity (resting pain), right pinch grip, and depression [49,50]. This difference is assumed to be due to the intervention period. While previous studies applied long-term exercise of 6–8 weeks, this study applied only a relatively short-term (three nights and four days) [49,50]. A previous study reported an improvement in grip strength of about 2.48 kg in cancer caregivers after 12 weeks of exercise [61]. Although this study and our study had environmental differences between the subjects, it showed that long-term exercise intervention was more effective than short-term exercise intervention in improving the caregiver’s strength (4 weeks: 0.3–0.6 kg vs. 12 weeks: 2.48 kg). 

There were significant time × group interactions on resting pain, movement pain, tactile sense, left grip strength, and physical health and environmental domains of QoL, with small (tactile sense: η^2^ = 0.11 in upper part, η^2^ = 0.19 in lower part; QoL: η^2^ = 0.24 in physical health, η^2^ = 0.24 in environmental domain), medium (left grip strength: η^2^ = 0.28, movement pain: η^2^ = 0.27), large (resting pain: η^2^ = 0.58) effect size. This difference is presumed to be due to the environmental characteristics of the urban region and marine regions environmental characteristics. The appreciation of natural scenery can promote cortisol reduction and serotonin secretion in the blood, improve mental and physical stability and emotional control, and allow the mind to relax [62]. These physical, psychological, and physiological changes could relieve musculoskeletal pain. Compared to the urban region, the subjects who performed the intervention in the marine region were exposed to many visual stimuli such as marine scenery, auditory stimuli such as the sound of the sea, and tactile stimuli such as sea sand and sea. The sea sand provides an unstable ground to the subject due to its mechanical properties, which require more physical muscle activity to maintain posture and perform movements [19]. In the body in contact with marine resources, sensory improvement can be promoted through more mechanical stimulation and proprioceptive sensory stimulation. In addition, natural ocean sounds can provide relaxation and improvement of mood to subjects as white noise [17]. Stimulation of these various sensory and motor nerves can further promote nerve activation through spatial summation and, together with it, can more effectively improve individual sensory perception. Similar to our results, Choi et al. reported that environmental factors were effective on subjects’ upper extremity muscle activity [63]. 

TSA showed a significant time × group interaction, and showed small (upper part: η^2^ = 0.11, lower part: η^2^ = 0.19) effect size. The MTG showed significant improvements at PI and FU, and showed large effect size in PI (upper part: *d* = 1.32, lower part: *d* = 1.11) and follow (upper part: *d* = 1.33, lower part: *d* = 0.92). The CG showed a tendency to improve TSA, but it was not significant. As the accuracy of the tactile sense is improved, the subject’s TSA value is lowered, which means that the subject’s ability to distinguish two nearby objects touching the skin is improved. This tactile sense can be reduced due to disease and aging, and a previous study suggested that prolonged mechanical noise improved the tactile sense [64]. We mentioned earlier that various sensory stimuli could be given to the subject in the marine region. In particular, it is presumed that multiple mechano-tactile stimuli such as sea sand, sea breeze, and seawater improved the tactile sense of the skin. In addition, pain relief by our intervention is considered a factor that improved TSA. Diseases and pain can induce changes in various functions of the body and the tactile sense [65,66]. Physiologically, pain can inhibit and reduce sensory transmission in surrounding tissues by lateral inhibition, interfering with the transmission of mechano-tactile sensation felt in the body surface area [67]. In this study, the pain was relieved through interventions such as appropriate exercise, meditation, and environmental factors such as the marine scenery. It is assumed that these physical benefits affected the improvement of TSA. 

In this study, the MTG showed significant improvement in physical health, psychological, environment, total of QoL, and depression at PI, with medium (psychological: *d* = 0.71) and large (physical health: *d* = 1.10, environment: *d* = 1.19, total: *d* = 0.93, depression: *d* = 0.89) effect size. The effect of intervention lasted for one month only in physical health, environment, and a total of QoL, and showed large (physiological health: *d* = 1.25, psychological: *d* = 0.84, environment: *d* = 1.25, total: *d* = 1.12, depression: *d* = 2.09) effect size. However, the CG showed significant improvement only in depression after the intervention with large (*d* = 1.23) effect size, and the effect of the intervention did not last. A study by Hughes (2013) reported that an 8-week meditation program had a positive effect on the stress of prehypertension participants [68], and several studies have reported that QoL and depression were improved through meditation [69,70,71]. In addition, there was a report that QoL was improved by applying the stabilization exercise intervention [45,72]. It is known that even short-term resistance exercises can reduce psychological discomfort [73]. These results are assumed to have also affected the reduction of pain. Pain is one of the factors that increase depression and lowers QoL by causing one complaint about discomfort [74,75]. In the results of this study, the pain was significantly reduced along with the improvement of QoL and depression. Like our results, previous studies were also accompanied by improvement in QoL, depression, and pain [76,77,78]. In addition, in the comparison between the MTG and CG, significant time × group interactions were shown in physical health, environmental, and a total of QoL. It is thought that the environment factor had a positive effect on QoL in addition to the effect of the program composed of meditation and the stabilization exercise. In a study by Kim and Koo (2019), it was reported that a healing program that included appreciation of the natural scenery improved depression and QoL of elderly persons living alone [79], and in a study by Lee et al. (2018) it was reported that mental-physical intervention using marine resources had a positive effect on QoL, psychology, and cognitive function [80]. For this reason, it is believed that the MTG had a positive effect on QoL and depression in this study and showed better results than the CG. 

This study suggests that meditation and physical exercise are effective for improving the physical and mental health of caregivers of people with disabilities, and it can also be seen that performing them in the marine region is more effective. Although marine therapy has a positive effect, it is necessary to move to a marine area to perform it. It is also accompanied by problems of travel time and cost. It may be difficult for caregivers to do this due to economic and environmental constraints. From the control group results in this study, we suggest that performing both interventions in a city could be an alternative method. It requires a relatively low budget and few restrictions on moving to a place and time. In addition, although it was not identified in our study, various sensory stimuli can be provided even in an urban environment through environmental manipulation, such as walking on sand or various sounds, and various positive effects can be expected through changes in meditation and exercise programs and additional programs such as painting. A clear effect and mechanism for this will be elucidated in future studies.

There are several limitations in this study. First, we applied a total of 8 sessions for 3 nights and 4 days, and this is the effect of a short-term intervention. Hence further research is required to clarify the long-term intervention effect. Second, the subjects who participated in this study were all women, so the effect of the marine healing program on male caregivers is still unclear. Third, the degree of physical activity in daily life other than the intervention could not be restricted. Fourth, the participant had an average care period of about 20 months, so it cannot be generalized for caregivers with longer care periods. Therefore, future studies that show the long-term effects on a wider range of caregivers such as care period and gender are needed.

## 5. Conclusions

This study demonstrated that meditation combined with stabilization exercise effectively improved pain, pinch grip-related muscle strength, and depression in female caregivers. In addition, it was confirmed that if these interventions were performed in the marine region, it was also effective in improving tactile sense, muscle strength related to handgrip, balance, and quality of life. In particular, the intervention performance in the marine region showed more significant improvement in pain and tactile sense than in the urban region. Although the same degree of intervention was applied to caregivers, there was a difference in effectiveness between the urban and marine regions. It is presumed that this was caused by providing more diverse sensory stimuli through white noise such as the sea sound and unstable ground such as sea sand. We suggest meditation and exercise intervention in the marine region to relieve the physical and mental burden and improve the quality of life of female caregivers.

## Figures and Tables

**Figure 1 ijerph-19-06025-f001:**
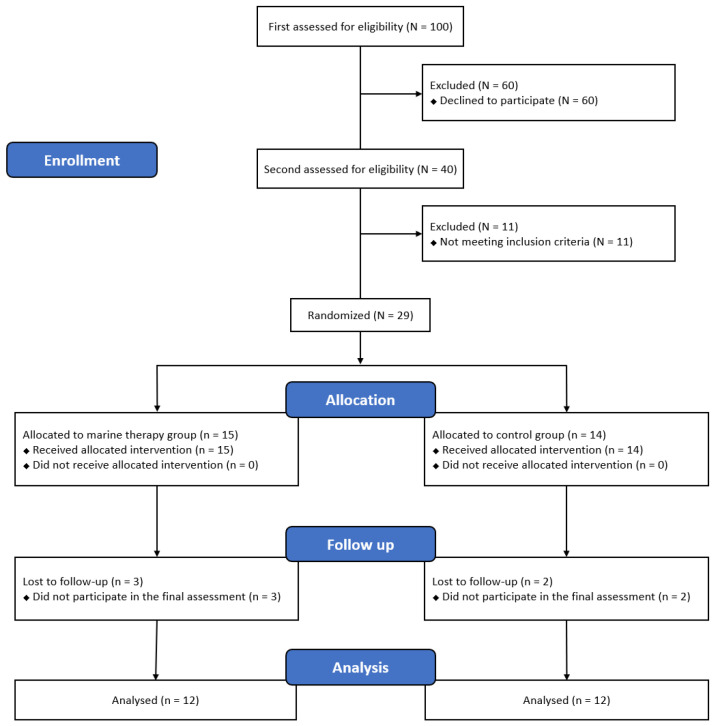
Experimental procedure.

**Table 1 ijerph-19-06025-t001:** General characteristics.

Variables		MTG (*n* = 12)	CG (*n* = 12)	t or Chi-Square (df)	*p*-Value
Age (year)		52.11 ± 10.34	53.40 ± 8.88	−0.292 (22) *	0.774
Height (cm)		157.21 ± 4.23	156.14 ± 6.18	0.436 (22) *	0.669
Weight (kg)		57.18 ± 8.68	61.12 ± 9.62	−0.934 (22) *	0.364
BMI (kg/m^2^)		23.14 ± 3.44	25.04 ± 3.41	−1.204 (22) *	0.245
Care period (month)		24.44 ± 25.02	15.90 ± 17.47	0.871 (22) *	0.396
Relationship					
	Wife	7 (58.30)	6 (50.00)	0.220 (3) ^#^	0.974
	Grandmother	1 (8.30)	1 (8.30)		
	Daughter	3 (25.00)	4 (33.30)		
	Daughter-in-law	1 (8.30)	1 (8.30)		

Continuous variables expressed as mean ± standard deviation, nominal variables expressed number (percentage). * using the *t*-test statistics; ^#^ using the chi-square test statistics. Abbreviation. MTG, marine therapy group; CG, control group; BMI, body mass index.

**Table 2 ijerph-19-06025-t002:** The changes of pain.

Variables	PRE	PI	FU	Source	F (df)	*p*	Partial η^2^
VAS (mm)							
Resting pain							
MTG	34.89 ± 9.96	10.11 ± 10.06 ^a,^*	13.78 ± 10.91 ^a,^*	Time	72.719 (1,22)	<0.001	0.81
CG	34.40 ± 6.79	25.50 ± 4.88 ^a^	33.50 ± 7.09	Group	10.487 (1,22)	0.005	0.38
d	0.37	1.45	1.64	T*G	23.184 (1,22)	<0.001	0.58
Movement pain							
MTG	51.33 ± 14.86	17.22 ± 9.72 ^a,^*	23.78 ± 7.40 ^a,^*	Time	24.952 (1,22)	<0.001	0.60
CG	46.40 ± 20.51	34.20 ± 17.74	39.10 ± 18.60	Group	2.250 (1,22)	0.152	0.12
d	0.24	1.07	1.02	T*G	6.384 (1,22)	0.004	0.27
PPT (kg)							
**UT**							
MTG	4.13 ± 2.29	4.67 ± 1.30	4.91 ± 1.46	Time	0.765 (1,22)	0.473	0.04
CG	5.08 ± 1.81	5.29 ± 2.80	5.40 ± 2.56	Group	0.686 (1,22)	0.419	0.04
d	0.36	0.26	0.21	T*G	0.136 (1,22)	0.873	0.01
**LS**							
MTG	3.92 ± 1.35	4.12 ± 1.04	4.89 ± 1.38	Time	0.213 (1,22)	0.809	0.01
CG	4.83 ± 1.21	4.68 ± 1.28	4.26 ± 1.40	Group	0.384 (1,22)	0.543	0.02
**d**	0.57	0.41	0.37	T*G	2.967 (1,22)	0.065	0.15
**SC**							
MTG	4.91 ± 2.22	4.89 ± 1.58	4.68 ± 1.05	Time	0.186 (1,22)	0.831	0.01
CG	5.15 ± 1.17	4.93 ± 1.78	4.92 ± 1.93	Group	0.076 (1,22)	0.786	0.00
d	0.10	0.02	0.14	T*G	0.046 (1,22)	0.955	0.00

Variables expressed the mean ± standard deviation. ^a^, *p* < 0.05 compared to PRE within groups; *, *p* < 0.05 compared to the CG in between the groups. Abbreviation. MTG, marine therapy group; CG, control group; PRE, pre intervention; PI, post intervention; FU, follow-up; VAS, visual analogue scale; PPT, pain pressure threshold; UT, upper trapezius; LS, levator scapulae; SC, splenius capitis; Cohen’s d, small ≥0.2, medium ≥0.5, large ≥0.8; η^2^, small ≥0.10, medium ≥0.25, large ≥0.40.

**Table 3 ijerph-19-06025-t003:** The changes of tactile sense.

Variables	PRE	PI	FU	Source	F (df)	*p*	Partial η^2^
Upper part (cm)							
MTG	3.58 ± 0.47	2.88 ± 0.34 ^a^	2.83 ± 0.43 ^a,^*	Time	14.460 (1,22)	<0.001	0.46
CG	3.51 ± 0.65	3.16 ± 0.25	3.21 ± 0.23	Group	1.898 (1,22)	0.186	0.10
d	0.10	0.74	0.82	T*G	2.169 (1,22)	0.130	0.11
Lower part (cm)							
MTG	8.39 ± 0.94	7.24 ± 0.61 ^a,^*	7.23 ± 1.20 ^a,^*	Time	7.672 (1,22)	0.002	0.31
CG	8.49 ± 0.88	8.25 ± 0.71	8.34 ± 0.69	Group	5.341 (1,22)	0.034	0.24
d	0.09	1.27	0.86	T*G	3.891 (1,22)	0.030	0.19

Variables expressed the mean ± standard deviation. ^a^, *p* < 0.05 compared to PRE in within groups; *, *p* < 0.05 compared to CG in between the groups. Abbreviation. MTG, marine therapy group; CG, control group; PRE, pre intervention; PI, post intervention; FU, follow-up. Cohen’s d, small ≥0.2, medium ≥0.5, large ≥0.8; η^2^, small ≥0.10, medium ≥0.25, large ≥0.40.

**Table 4 ijerph-19-06025-t004:** The changes of muscle characteristics.

Variables	PRE	PI	FU	Source	F (df)	*p*	Partial η^2^
**UT stiffness (N/m)**							
MTG	237.33 ± 29.74	223.11 ± 28.14	230.33 ± 18.46	Time	1.486 (1,22)	0.241	0.08
CG	239.10 ± 22.42	240.20 ± 27.48	231.40 ± 20.83	Group	0.438 (1,22)	0.517	0.03
d	0.05	0.50	0.05	T*G	1.871 (1,22)	0.170	0.10
**UT Elasticity (logarithm)**							
MTG	1.92 ± 0.35	1.84 ± 0.34	1.94 ± 0.28	Time	0.358 (1,22)	0.702	0.02
CG	2.00 ± 0.58	1.91 ± 0.51	1.90 ± 0.44	Group	0.044 (1,22)	0.837	0.00
d	0.14	0.13	0.11	T*G	0.195 (1,22)	0.824	0.01
**LS stiffness (N/m)**							
MTG	239.22 ± 31.16	240.00 ± 31.18	251.11 ± 25.48	Time	2.440 (1,22)	0.102	0.13
CG	236.70 ± 18.57	243.60 ± 21.51	257.80 ± 35.89	Group	0.081 (1,22)	0.779	0.01
d	0.07	0.10	0.19	T*G	0.180 (1,22)	0.836	0.01
**LS Elasticity (logarithm)**							
MTG	1.52 ± 0.22	1.50 ± 0.27	1.61 ± 0.25	Time	0.544 (1,22)	0.585	0.03
CG	1.73 ± 0.24	1.70 ± 0.23	1.71 ± 0.23	Group	3.481 (1,22)	0.079	0.17
d	0.74	0.64	0.34	T*G	0.580 (1,22)	0.565	0.03
**SC stiffness (N/m)**							
MTG	340.44 ± 63.91	366.22 ± 32.56	354.11 ± 45.43	Time	5.814 (1,22)	0.007	0.26
CG	342.90 ± 31.90	381.50 ± 57.75	395.40 ± 51.53	Group	1.149 (1,22)	0.299	0.06
d	0.04	0.29	0.71	T*G	1.602 (1,22)	0.216	0.09
**SC Elasticity (logarithm)**							
MTG	1.46 ± 0.12	1.47 ± 0.24	1.35 ± 0.22	Time	6.707 (1,22)	0.004	0.28
CG	1.54 ± 0.11	1.53 ± 0.16 ^a^	1.38 ± 0.19	Group	0.652 (1,22)	0.431	0.04
d	0.54	0.22	0.08	T*G	0.225 (1,22)	0.799	0.01

Variables expressed the mean ± standard deviation. ^a^, *p* < 0.05 compared to PRE in within groups. Abbreviation. MTG, marine therapy group; CG, control group; PRE, pre intervention; PI, post intervention; FU, follow-up; UT, upper trapezius; LS, levator scapula; SC, splenius capitis; Cohen’s d, small ≥0.2, medium ≥0.5, large ≥0.8; η^2^, small ≥0.10, medium ≥0.25, large ≥0.40.

**Table 5 ijerph-19-06025-t005:** The changes of muscle strength and balance.

Variables	PRE	PI	FU	Source	F (df)	*p*	Partial η^2^
**Muscle strength (kg)**							
**Grip strength (Rt)**							
MTG	22.67 ± 2.77	24.28 ± 3.30 ^a^	24.06 ± 3.57 ^a^	Time	5.439 (1,22)	0.009	0.24
CG	22.89 ± 5.13	23.51 ± 5.06	23.04 ± 5.27	Group	0.070 (1,22)	0.794	0.00
d	0.05	0.16	0.20	T*G	1.792 (1,22)	0.182	0.10
**Grip strength (Lt)**							
MTG	19.66 ± 3.76	21.95 ± 3.77 ^a^	21.84 ± 3.77 ^a^	Time	5.979 (1,22)	0.006	0.26
CG	21.81 ± 6.73	22.17 ± 5.92	21.24 ± 5.83	Group	0.064 (1,22)	0.803	0.00
d	0.36	0.04	0.11	T*G	6.711 (1,22)	0.003	0.28
**Pinch grip (Rt)**							
MTG	6.62 ± 0.71 ^a^	7.50 ± 0.73	6.71 ± 1.02	Time	15.265 (1,22)	0.000	0.47
CG	6.51 ± 0.99 ^a^	7.28 ± 1.57	6.66 ± 1.27	Group	0.073 (1,22)	0.790	0.00
d	0.11	0.17	0.04	T*G	0.141 (1,22)	0.869	0.01
**Pinch grip (Lt)**							
MTG	6.29 ± 0.41 ^a^	6.86 ± 0.65	6.76 ± 0.65	Time	4.234 (1,22)	0.023	0.20
CG	6.19 ± 0.92	6.21 ± 0.96	6.36 ± 0.98	Group	1.255 (1,22)	0.278	0.07
d	0.13	0.69	0.42	T*G	2.530 (1,22)	0.095	0.13
**Timed up and go (s)**							
MTG	8.09 ± 1.17 ^a^	7.63 ± 0.96	7.38 ± 0.90	Time	4.196 (1,22)	0.024	0.20
CG	8.02 ± 0.86	7.63 ± 0.55	7.79 ± 0.75	Group	0.103 (1,22)	0.752	0.01
d	0.06	0.00	0.39	T*G	1.069 (1,22)	0.355	0.06

Variables expressed the mean ± standard deviation. ^a^, *p* < 0.05 compared to PRE in within groups. Abbreviation. MTG, marine therapy group; CG, control group; PRE, pre intervention; PI, post intervention; FU, follow-up; Cohen’s d, small ≥0.2, medium ≥0.5, large ≥0.8; η^2^, small ≥0.10, medium ≥0.25, large ≥0.40.

**Table 6 ijerph-19-06025-t006:** The changes of quality of life and depression.

Variables	PRE	PI	FU	Source	F (df)	*p*	Partial η^2^
**K-WHOQOL-BREF (score)**							
**Overall QOL & GH**							
MTG	53.33 ± 15.81	65.56 ± 19.44	70.00 ± 14.14 ^a^	Time	5.443 (1,22)	0.009	0.24
CG	58.00 ± 4.22	63.00 ± 4.83	61.00 ± 5.68	Group	0.328 (1,22)	0.574	0.02
d	0.29	0.13	0.61	T*G	2.215 (1,22)	0.125	0.12
**Physical health**							
MTG	50.48 ± 12.29	68.25 ± 14.81 ^a^	68.89 ± 11.38 ^a^	Time	13.991 (1,22)	< 0.001	0.45
CG	59.43 ± 5.99	68.86 ± 8.13	60.57 ± 9.11	Group	0.012 (1,22)	0.914	0.00
d	0.69	0.04	0.64	T*G	5.302 (1,22)	0.010	0.24
**Psychological**							
MTG	51.85 ± 13.45	65.18 ± 18.72 ^a^	65.93 ± 14.02 ^a^	Time	9.946 (1,22)	< 0.001	0.37
CG	55.67 ± 8.32	65.67 ± 8.47	60.00 ± 10.18	Group	0.013 (1,22)	0.911	0.00
d	0.23	0.04	0.42	T*G	1.611 (1,22)	0.215	0.09
**Social relationships**							
MTG	51.85 ± 13.45	65.18 ± 18.72	65.93 ± 14.02	Time	1.051 (1,22)	0.361	0.06
CG	55.67 ± 8.32	65.67 ± 8.47	60.00 ± 10.18	Group	1.217 (1,22)	0.285	0.07
d	0.05	0.36	0.53	T*G	0.535 (1,22)	0.591	0.03
**Environmental**							
MTG	48.61 ± 9.36	62.78 ± 10.42 ^a^	63.06 ± 9.66 ^a^	Time	20.004 (1,22)	< 0.001	0.54
CG	55.25 ± 9.24	59.50 ± 12.35	60.25 ± 13.51	Group	0.002 (1,22)	0.968	0.00
d	0.53	0.22	0.20	T*G	5.229 (1,22)	0.010	0.24
**Total**							
MTG	53.30 ± 10.48	66.13 ± 12.71 ^a^	67.50 ± 10.03 ^a^	Time	11.958 (1,22)	< 0.001	0.41
CG	58.07 ± 3.66	64.20 ± 5.22	60.90 ± 6.10	Group	72.719 (1,22)	< 0.001	0.81
d	0.42	0.12	0.63	T*G	24.952 (1,22)	< 0.001	0.60
**K-HDRS (score)**							
MTG	17.56 ± 7.35	11.00 ± 1.22 ^a^	11.89 ± 2.03	Time	13.312 (1,22)	< 0.001	0.44
CG	17.60 ± 4.20	12.20 ± 1.75 ^a^	13.00 ± 1.94	Group	0.781 (1,22)	0.389	0.04
d	0.01	0.69	0.45	T*G	0.131 (1,22)	0.877	0.01

Variables expressed the mean ± standard deviation. ^a^, *p* < 0.05 compared to PRE in within groups. Abbreviation. MTG, marine therapy group; CG, control group; PRE, pre intervention; PI, post intervention; FU, follow-up, K-WHOQOL-BREF, Korean version of world health organization quality of life instrument-short version; QOL & GH, quality of life and general health; K-HDRS, Korean version-Hamilton Depression Rating Scale; Cohen’s d, small ≥0.2, medium ≥0.5, large ≥0.8; η^2^, small ≥0.10, medium ≥0.25, large ≥0.40.

## Data Availability

The original contributions presented in the study are in this article; further inquiries should be directed to the corresponding authors.

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
