# Peer review of "The Effects of Meditation with Stabilization Exercise in Marine Region on Pain, Tactile Sense, Muscle Characteristics and Strength, Balance, Quality of Life, and Depression in Female Family Caregivers of People with Severe Physical Disabilities: A Randomized Single-Blinded Controlled Pilot Study"

_ijerph, 2022, doi:10.3390/ijerph19106025_

Round 1

Reviewer 1 Report

Thank you for giving me the opportunity to read your manuscript “The Effects of Meditation with Stabilization Exercise on Mental Health, Sensory Function, Muscle Characteristics, and Physical Health in Female Caregivers of Patients with Severe Physical Disabilities in Marine Region.”

I have some concern and suggestions how the manuscript could be improved. My main concerns are summarized and followed by detailed suggestions:

I) Report effect sizes.

II) Explain why scenery should make such a difference – especially in pain.

III) The effect of the intervention at the sea side, focuses on how the scenery might help. However, in the Discussion it becomes clear that participants at the sea side also had a change in the pace of their day and seem to have exercised more. This same conditions seem not to apply to the control group. This free time and exercise (or doing something that is fun in general) might have made the difference – and might have nothing to do with the scenery or the place fun was experienced (could have been a park or amusement park in the city as well – maybe)

My detailed concerns and suggestions:

Major Points

Overall

1) Be very careful with language. Try to use bias-free formulations. Thus, try to use the “persons first approach” (https://apastyle.apa.org/style-grammar-guidelines/bias-free-language/disability) and write “patients with disability” instead of “disabled patients”.

2) In how far are people for whom is being cared for “patients”? Maybe “people with disability”

Abstract

2) If you report the p value in the Abstract (e.g., “p < .05”) first, write the exact p value (if is not below .001), second, report the test statistics as well, i.e., t-test and degrees of freedom.

3) In order to see how large of the intervention was, effect sizes (e.g., Cohens d) would also be interesting.

4) It seems that both interventions help; but that marine might also address more parameters of quality of life. In the Conclusions of the Abstract I would suggests mentioning that both interventions seem to have some effect.

Introduction

5) In the first sentence there is the statement that number of long-term inpatients increases. However, references 1 and 2 do not report about increasing numbers. I suggest either using a reference for this statement, or to start that most share of the care for family members with disability is informal. (There the transition that those caregivers are mostly women might fit. You can explain and mention how traditional gender stereotypes and resulting share of work results that women are often those who perform caregiving obligations.)

6) I would suggest “improve” (or similar terms) instead of “heal” (e.g., line 50). Heal seems a too strong word.

7) Remove “even” from “The effects of these interventions on QoL, stress, depression, and physical burden even for caregivers have been reported by several researchers” – the study and the focus was on caregivers.

8) Offer references for “In particular, it has been reported that elements of marine resources such as seawater, marine sand, sea breezes, and marine scenery have a positive effect on mental and physical health.”

9) “Using the Dead Sea” (line 58) is a confusing expression to me. Maybe “at the Dead See shore” or “climatotherapy at the Dead Sea” or “bath in the Dead Sea” – in the expression “using the Dead Sea” it does not seem clear to me what was done with the Dead Sea.

10) Report and explain that you are only going to study women (see Comment 5).

11) Shortly try to explain how the interventions work, and why it should have effect on muscle strength, muscle characteristics, and musculoskeletal pain. Mood or emotions are different from the experience of pain – especially if the body is strained by repeated unhealthy movement.

Methods

12) What patients (line 68). It seemed that you are going to study caregivers – not people with disabilities whom you called “patients” so far.

13) Give more information about the participants. I would move “General Characteristics of Participants” to the “Participants” section of the Methods.

14) Report the test statistics (t(df) = ; or Chi Square (df) = ) in Table 1.

15) Give some calculations on expected sample size and use effect size estimates from previous studies.

16) Report some participant characteristics in the main text.

17) In the procedures also report how you contacted or recruited your participants.

18) Who and how was caregivers “subjective care stress” confirmed? Did participants have a diagnosis? Who were the experts? (line 74)

19) I would change the Title of Figure 1 that reads “CONSORT flow chart” – maybe “participants”

20) Avoid using “etc.” (line 119) What exactly was assessed?

21) Use references for “Time up and go (TUG)” lines 160-167

22) Report some psychometric properties of QoL and Depression scales – at least original Cronbach Alphas and own Cronbach Alphas

23) Report the source for the Korean version of the WHOQOL-BREF as well. Or if you translated it – how did you do the translation?

24) Report the poles of the Likert scale “1 = …; 5 = …”; also give if possible an example item (for each scale you analyzed).

25) Why were points transformed? (line 175) – if you do the transformation cite the source again.

26) Similar points of concern for the K-HDRS as for the QoL (see Comments 22 – 24)

27) Offer some definition for the psychological construct QoL and depression.

28) Why was this interval and this frequency of the application of the intervention chosen? (line 190)

29) Maybe order the chapters in the same order as you mention the exercises in “The intervention consisted of meditation and a stabilization exercise to promote physical activity”

30) Report effect sizes (Cohens d, Cramers V, η2)

31) I suggest you define what conditions (before, PI, FU) you are going to contrast. Thus, you can define a-priori contrasts. Are you comparing “before vs. PI” and then “PI vs. FU”; or “before vs. PI” and then “before vs. FU”? Add this information to the statistical analysis. You should only do two comparisons – not three.

Results

32) In relation to Comment 31; Because it is not clear which groups you compared it is not clear what you mean by “whereas in the CG, resting pain showed a significant decrease only at PI (p < .05), which did not last” – is “before vs. FU” non significant?

33) As stated for the Abstract the p values in the main text do add little information. Better refer to Table 2 earlier in the text.

34) Report the overall F statistics in Table 2 and also the test statistics for all the group comparison tests. Report effect sizes.

35) Better report p values “1.000” as “> 0.999”

36) Was the Bonferonni correction applied to the whole study (i.e., was p adjusted to all the comparisons you did; or only to post-hoc tests).

37) If you adjusted the p-value for 3 comparisons – thus set it to 0.017, the reported difference in e.g., grip strength right hand (p = 0.042) does not seem significant. Also General Health or Depression does not seem to change significantly.

38) I would avoid writing “Interestingly, …” the interaction was significant. But rather write, that the significant interaction term indicates that the groups differed in the change over time, as described.

Discussion

39) I do not understand what is meant by “and the following results were confirmed” line 305 – I suggest deleting.

40) in line 330 you explain that “greater improvement was due to the marine scenery and meditation interventions” – however, the control group also did meditation (according to line 92; “while CG performed the same interventions”).

41) “However, compared to previous studies” – which studies?

42) “The leisure time of MTG was spent walking on a beach and viewing the marine scenery.” It seems that the MTG was on some kind of vacation. Was the CG also on “vacation” and freed from caregiver obligations during those days? Did the CG also stay at a hotel or did they go home after the intervention?

43) Why should “various marine resources promote[] the process of awareness and acceptance of the participant, and increased mood, cognitive flexibility, and efficiency” ?

44) Maybe be more descriptive what you mean with “TSA” in the discussion (line 350)

45) Give references for the explanation of horizontal inhibition (line 353)

46) “recreational activities had a positive effect on QoL in addition to the effect of the program” line 375 – what recreational activities? This seems to be important – and might to have nothing to do with the scenery. Thus, the control group and the intervention group seem to differ in other key aspect. The control group seems to have had free time and spend time away from home (and thus away from caregiver obligation) whereas we get to know little about the control group. This needs to be clarified, because it seems that not the “intervention”, i.e., sea side was the key factor but the surrounding circumstances (free time, exercises at the beach, having free time; or simply fun).

47) Try to offer some suggestion already in the Discussion. There you could also discuss advantages and disadvantages of the chosen locations. E.g., needing more time and spending time away from home, for beach side interventions. Also costs. And maybe also stress more that city location interventions had positive effects.

48) If city location interventions had so little positive effects, maybe the intervention could be modified and e.g., mediation could be removed? Maybe a sports program and more exercises can be suggested?

Minor:

Some typos in lines 68-82. E.g., “The participant who participated” – “participants”; “those who complains of pain” – “complain” “those who has experienced” – “have”

Author Response

  • Manuscript Number: ijerph-1694134
  • Manuscript Title: The Effects of Meditation with Stabilization Exercise on Pain, Tactile sense, Muscle Characteristics and Strength, Balance, Quality of Life, and Depression in Female Caregivers of Patients with Severe Physical Disabilities in Marine Region: A Randomized Single-Blinded Controlled Pilot Study
  • Type of work: Article

Dear Reviewers,

I would like to thank you for providing the opportunity to revise and resubmit the attached manuscript entitled “The Effects of Meditation with Stabilization Exercise on Pain, Tactile sense, Muscle Characteristics and Strength, Balance, Quality of Life, and Depression in Female Caregivers of Patients with Severe Physical Disabilities in Marine Region: A Randomized Single-Blinded Controlled Pilot Study” for publication in IJERPH.

We deeply appreciate the editorial comments and reviewers’ helpful comments on our manuscript which we ignored. We agreed with the points addressed by the Reviewers. We provide our responses to the Reviewers’ comments. Please review the attached files.

Reviewer 2 Report

I read with interest the study entitled "The Effects of Meditation with Stabilization Exercise on Mental Health, Sensory Function, Muscle Characteristics, and Physical Health in Female Caregivers of Patients with Severe Physical Disabilities in the Marine Region". The study has an interesting topic but the good news stops there. The main problem of the study is the small population of the sample in relation to the many factors it considers (only the quality of life includes 5 subscales). The following are my other remarks:

Introduction: The introduction does not provide a sufficient background, the references related to meditation are minimal.

Materials and Methods: Missing data for calculating the minimal required sample.

Results:

- The authors do not report the results from the normal distribution test (Shapiro-Wilk test).

- Table 1: Marital status and Relationship do not describe all participants.

I would also like to point out that the study does not examine whether the results may be due to the four-day sea vacation regardless of meditation as there is no corresponding control group.

In my conclusion, I would suggest to the authors to withdraw the paper and rewrite it in the form of a pilot study.

Author Response

(The authors gave the same response as above.)

Reviewer 3 Report

Thank you for the opportunity to review the paper entitled “The Effects of Meditation with Stabilization Exercise on Mental Health, Sensory Function, Muscle Characteristics, and Physical Health in Female Caregivers of Patients with Severe Physical Disabilities in Marine Region”. The intervention of applying natural marine resources for physical health such as muscle strength, muscle characteristics, and musculoskeletal pain in caregivers is very innovative. Below are a few comments:    

  1. Since one of the inclusion criteria is “non-specific back pain and neck pain”, could you please clarify what the authors refer to by “non-specific”.
  2. Are there any incentives for the participants?
  3. The intervention was performed 8 times, with 2 sessions per day for 4 days and 3 nights and 60 minutes per session, suggesting that the participants only received the intervention for 2 hours per day. Did the authors ask the participants what they did for the remaining time? This is important information as the participants may walk on the beach etc which contributed to some physical exercises. This may also contaminate the data.
  4. Another limitation of the study is that the sample size is small.

Author Response

(The authors gave the same response as above.)

Reviewer 4 Report

Dear authors, 

Basically, the research is decent but requires for sure certain improvements.

So, these are my comments and recommendations:

  • the article needs to have some objectives (here in the Introduction section could be declared some paper goal/goals);
  • also is madatory to have a special section called Literature review, even though in the Introduction are mentioned some papers or studies;
  • in terms of participation I need to know, and you have to explain several issues: who decided the inclusion and exclusion criteria and based on what?; Increased number for exclusion criteria does not introduce increased selectivity for the study? Also, for the third inclusion criteria could you give more details, explanations? When were carried out these type of interviews, based on what, in what context etc.? Also, the sessions applied (as interventions) for the two groups (MTG and CG) were carried out by the same people? Or these sessions are carried out in parallel? It was a training session for the people involved in the study? How long did it last and what did it consist of?
  • Did the authors have a discussion with the therapists that are working in these medical facilities in order to build the appropriate and objective procedure? Were certain aspects highlighted on this occasion that need special attention? Are there differences in approach for the two types of therapies in relation to the conditions of the interventions?
  • The Conclusions section is poor and vague constructed. In principle, if there is a prior discussion with specialists in the field, more than likely the idea included here was the same. However, what is the original, novel part that the study proposes? What is really remarkable and can be introduced here? A clearer emphasis is needed/required to underline the usefulness of the study.

I ask the authors to pay attention to all the comments in responding to these observations and to include pertinent comments, which are absolutely necessary for readers interested in the field.

Author Response

(The authors gave the same response as above.)

Reviewer 5 Report

Dear Authors, I have read your manuscript with interest.

The current manuscript titled: "The Effects of Meditation with Stabilization Exercise on Mental Health, Sensory Function, Muscle Characteristics, and Physical Health in Female Caregivers of Patients with Severe Physical Disabilities in Marine Region" represents an important analysis of evolving field of Psychology and Public Health.

In my opinion, the number of patients included in this study is insufficient and cannot reflect reliable statistical results and, moreover, conclusions. In addition, the time interval of the patients study is very short and errors in the results are not excluded for this reason. The study methodology is not complex, so there is no objective reason for the small number of patients included. I believe that this results volume does not correspond to the scientific level of this Journal, the reason for which I advise to increase the number of patients (or increase the time interval for examining patients) and revise the study results or consider the possibility of publishing the results in a journal with a lower impact factor.

Also, these are the adjustments which should be made to increase the value of your future manuscript submission:

  1. Line 21: please, add abbreviation for MTG and CG.
  2. Line 38: dementia is also a psychiatric disease, so I advise you to change “neurological diseases” to ”psycho-neurological diseases”.
  3. In the Introduction section, please add information about the scientifically proven physiopathological mechanisms of the influence of yoga / meditation on the human body. This information will give your article more scientific value and help readers better understand the study results.
  4. Line 71: what criteria was used to select this age range?
  5. In Figure 1, I advice to delete in the Excluded patients, information which include N=0. This information has no practical value.
  6. Line 115: change please “cm2” to “cm2”.
  7. Table 1: why information about Relationship is important? Explain please.
  8. In the Discussions section, there is a lot of information that repeats the Results section. Please try to comment on the results obtained in this study and compare them with the results of other authors.

Good luck!

Author Response

(The authors gave the same response as above.)

Round 2

Reviewer 1 Report

I commend the authors for the extensive changed they made to their original version of the manuscript. I think the manuscript has improved a lot. Especially, the rationale why the intervention should be different in the city and the shoreline has improved.

Some concerns remained. I have responded to authors’ response in cases concerns remained.

  Comments from Reviewer 1:

  1. Explain why scenery should make such a difference – especially in pain.

Response:

  • We deeply appreciate your good comments. In response to your point, we have added the following in the Introduction section.

  • In line 71-75: “However, it has been reported that meditation performed in the forest or wild nature produces more positive effects on improvement of pain, depression, stress, and QoL than conducted in a garden or park [16]. These results indicate that the environment can be an important factor in the subject's sensory function and psychological changes.”

>>> Thank you for adding this important reference. First I would mention that this is a meta-analysis. Saying, that it “has been reported” sound as though you rely here on only one study – which is not the case.
Second, try maybe to use the theory summarized in this review-paper (e.g., point 1.3; page 2). There Kaplan’s attention restoration theory (ART) is mentioned. Having a theoretical background, that argues why environmental exposure should be help-full would give much weight to your argument and intervention.

Major point

Overall

  1. In how far are people for whom is being cared for “patients”? Maybe “people with disability”

Response:

  • The subjects of this study were family caregivers caring for people with disabilities with stroke or traumatic brain injury who need support to perform daily activities. To express this more clearly, we revised the inclusion criteria of the participants as follows.

  • In line 105-111: “The inclusion criteria were as follows: 1) aged > 19 and < 65, a female caregiver who cares for patients with severe physical disability (e.g., stroke (over moderate ≥ 9 NIHSS (National Institutes of Health Stroke Scale), or traumatic brain injury) [21,22], 2) non-specific low back pain and neck pain in which a neurologist detected no neuropathological problem or structural damage, and 3) who has confirmed subjective stress due to caring through an interview with one psychiatrist (HDRS score ≥ 8) [23]”

>>> I wanted to put emphasis on the fact that not all people with disabilities are “patients” – also those who need assistance in everyday living. Thus, I would (re-)consider whether calling “people with disability” “patients” is appropriate or justified.

Are people with disability “inpatients” thus staying/living at the hospital? (Methods, first sentence)

>>> Additionally, I do not understand the new term “family caregiver.” For whom else is the participant looking after?

Abstract

  1. If you report the p value in the Abstract (e.g., “p < .05”) first, write the exact p value (if is not below .001), second, report the test statistics as well, i.e., t-test and degrees of freedom.

Response:

  • We present the exact p-value and degrees of freedom in the Abstract section and Tables according to your suggestion.

>>> Please also report the test statistics, e.g., t(1) = 2.1, p = 0.002

Introduction

  • In addition, we have additionally described female family caregivers, along with appropriate references.

  • In line 56-64: “The social roles of women and men have also changed along with the changes in society and industry. Still, in Korea, the socio-cultural trend continues in which men are responsible for economic support and women take on various household chores. Along with this trend, the role of caring for a family with physical or mental disabilities is mainly taken by women, and many women have been experiencing physical or mental burdens due to long-term care [9,10]. Unfortunately, however, most studies to date have focused on treatment or symptom improvement methods for people with disability, and studies on the health promotion of their family caregivers are very scarce.”

>>> I think this is an important addition, because it explains, why you only chose women to participate. Please start this section with a new paragraph.

I would not call it “trend” but “gender specific norms” or “traditional work load allocation”

  1. Shortly try to explain how the interventions work, and why it should have effect on muscle strength, muscle characteristics, and musculoskeletal pain. Mood or emotions are different from the experience of pain – especially if the body is strained by repeated unhealthy movement.

Response:

  • Before designing this study, we investigated the body's most uncomfortable points and mental stress for family caregivers. We confirmed that the main inconveniences were physical strain along with pain. We added a new reference for this and also described the contents in addition to the Introduction.
  • We performed the same therapeutic exercise in sea sand and city in patients with chronic ankle instability and reported the results. Interestingly, it improved pain and balance when performing exercises on sand compared to those performed on stable urban surfaces. We designed this study based on these experiences, and the contents were additionally described in the Introduction.
  • In your opinion, mood and emotion are different from pain. Of course, these are related in the case of various diseases accompanied by chronic pain or a condition accompanied by extreme pain. Still, I agree that the essence of these is obviously different. Our study individually confirmed the effects of meditation and stabilization exercise on measurement outcomes but did not analyze the correlation between them.

  • In line 48-51: “The family caregiver who takes care of long-term physical disability experiences mental stress due to lack of sleep, insufficient exercise, and absence of rest, and also has a physical burden (fatigue, backache, physical strain) due to the frequent patient transfer and uncomfortable postures of the caregivers [4-7].”

>>> Please use the „person first“approach – thus, “caregiver who takes care of people with long-term physical disability”

  • In line 85-88: “Our previous study confirmed that therapeutic exercise in sea sands was effective in reducing pain and improving motor function in patients with chronic ankle instability, suggesting that these effects appear through various sensory stimuli and activation of proprioceptive sensations in the feet and body [19].”

>>> Similar to the first comment; it seems important or would add strong arguments for your study, if you explained why such an intervention might address or improve experiences of pain. Using other empirical findings (as you did) is the second step; before that I would suggest adding a short theoretical explanation.

>>> I would suggest stating your second hypothesis in a similar manner as you did the first hypothesis. Thus, again use terms “investigate” or “find out” or “study”. “Demonstrate” seems too pre-defined to me.

Methods

  1. What patients (line 68). It seemed that you are going to study caregivers – not people with disabilities whom you called “patients” so far.

Response:

  • We modified “participants” to “female caregivers”, and “patient” to “those who”.
  • In line 102-104: “Forty female family caregivers volunteered to participate in this study, and 24 subjects finally participated in consideration of the inclusion and exclusion criteria.”

>>> Thank you for addressing this point. I think I was confused by the term “patient”. “Participants” is a good expression; I do not think that “family caregivers” is making it easier to understand. I would stick to “caregivers” for people with disabilities.

  • In line 113-115: “2) those who are accompanied by specific back pain that appears due to causes such as nerve root pain, infection, spinal cord injury, trauma, cauda equina syndrome, and structural deformation”

>>> “The exclusion criteria were as follows” maybe better “Excluded from the study were female caregivers with the following characteristics: …”

  1. Report the test statistics (t(df) = ; or Chi Square (df) = ) in Table 1.

Response:

  • We have provided additional information about (df) in the Tables based on your point.

  1. Report some psychometric properties of QoL and Depression scales – at least original Cronbach Alphas and own Cronbach Alphas

Response:

  • As in the case of TUG, we additionally provided Cronbach's alpha information for measurements of QoL and depression.
  • In line 234-236: “QoL was measured using the Korean version of the World Health Organization quality of life instrument-short version (K-WHOQOL-BREF), with a Cronbach's alpha of 0.898 [34,35].”
  • In line 247-248: “Depression Rating Scale (K-HDRS), which has Cronbach’s alpha of 0.76 [36,37].”

>>> Could you also report Cronbach’s alphas from your own study?

  1. Report the poles of the Likert scale “1 = …; 5 = …”; also give if possible an example item (for each scale you analyzed).

Response:

  • We presented that the score of the measurement method was converted to 100 points and described the meaning of high scores.
  • In line 241-245: “. The score for each domain was converted to a perfect score of 100 [34]. Questions 3, 4, and 26 that are related to being positive were inverted and summed with the other questions to be scored. The total score was converted into 100 points for each domain. A higher score means that the perception or satisfaction of the individual for the corresponding domain is increased.”

>>> When you state that responses are given on a Likert-scale it would be informative to know how the poles of the scale were labeled. E.g. “1 = totally disagress, 5 = totally agree”. Furthermore, it would be interesting if you gave on example item per scale – so that readers can imagine what exactly was asked.

  1. Offer some definition for the psychological construct QoL and depression.

Response:

  • We sincerely appreciate your good comments. According to WHOQOL's user manual, this measure is presented as a program for mental health. Also, as you know, QoL includes a physical health domain and a mental health domain. As this measure consists of both domains, it has positive and negative aspects to each other's measures. We wanted to characterize the various changes in caregivers, so we performed several measurements. In summarizing this information, considering the opinions of psychiatrists and clinical psychologists, QoL and depression were classified as mental health. In addition, the previous studies that thought of them as the realm of mental health were also considered.
    • WHOQOL user manual: https://apps.who.int/iris/handle/10665/63482
    • QOL & Mental health: https://www.ncbi.nlm.nih.gov/pmc/articles/PMC4224500/
    • Depression & Mental health: https://pubmed.ncbi.nlm.nih.gov/31987241/
  • However, we respect your opinion and describe the measurement variables separately.

>>> So far Quality of Life and other concepts have not been defined. It is important that the concepts that were measured are defined. Think about stating an example item to increase clarity.

  1. 30) Report effect sizes (Cohens d, Cramers V, η2)

Response:

  • We present the effect size in Tables 2-6 according to your suggestion.

>>> Report how to interpret the effect sizes – by citing a reference

Later also include the information in your Discussion.

  1. I suggest you define what conditions (before, PI, FU) you are going to contrast. Thus, you can define a-priori contrasts. Are you comparing “before vs. PI” and then “PI vs. FU”; or “before vs. PI” and then “before vs. FU”? Add this information to the statistical analysis. You should only do two comparisons – not three.

Response:

  • Thanks to all authors for your good comments. We have further described this in the Statistical analysis section.
  • In line 286-289: “Repeated-measures ANOVA was used to analyze the changes in variables between the groups (MTG, CG) over time (PRE, PI, FU). For the post-hoc analysis, the independent t-test was used to compare two groups, and the paired t-test was used to confirm the difference over time (PRE, PI, FU) within the group.”

>>> You should do only two comparisons – either [“before vs. PI” and then “PI vs. FU”] or [“before vs. PI” and then “before vs. FU”]. Do not perform three comparisons – thus, do not compare all conditions (there is no degree of freedom left in such a case); You should only do two comparisons – not three.

Additionally, because you have a clear hypothesis, you can define “contrasts” in you ANOVA analysis. You do not have to do “post-hoc” tests.

Result

  1. In relation to Comment 31; Because it is not clear which groups you compared it is not clear what you mean by “whereas in the CG, resting pain showed a significant decrease only at PI (p < .05), which did not last” – is “before vs. FU” non significant?

Response:

  • We agree with your opinion. So we added the following:
  • In line 293-295: “In the MTG, resting pain and movement pain both showed a significant decrease at PI and FU (p < .05), whereas in the CG, resting pain showed a significant decrease at PI (p < .05) but significantly increased pain in FU compared to PI (p > .05).”

>>> “Compared to PRE, in the MTG, resting pain … “

I suggest comparing [“before vs. PI” and then “before vs. FU”]

  1. Report the overall F statistics in Table 2 and also the test statistics for all the group comparison tests. Report effect sizes.

Response:

  • According to your suggestion, we have presented F values and effect sizes in Tables 2 to 6.

>>> Much better; please also report degrees of freedom for every analysis.

  1. Better report p values “1.000” as “> 0.999”

Response:

  • Thank you very much for your attentive advice. As per your point, we have marked the p-value.
  • In the Table 2-6

>>> Please take one more look at all tables. I think Table 3 first line p is stated to be 0.000

  1. Was the Bonferroni correction applied to the whole study (i.e., was p adjusted to all the comparisons you did; or only to post-hoc tests).

Response:

  • We applied the Bonferroni correction to the analysis of all measurement variables. First of all, I would like to apologize for not writing this in the manuscript. We present this in the Statistics analysis section. All authors are grateful for your good comments.
  • In line 289-290: “The Bonferroni correction was applied to prevent Type 1 error.”
  1. If you adjusted the p-value for 3 comparisons – thus set it to 0.017, the reported difference in e.g., grip strength right hand (p = 0.042) does not seem significant. Also General Health or Depression does not seem to change significantly.

Response:

  • Bonferroni correction was applied to all outcome measurements, and the p-value presented in the table was not corrected from 0.05 to 0.017, but the p-value obtained from the result value was x3 applied through Bonferroni correction.

>>> If the Boferonni correction was applied then also describe how. How many comparisons were planned? And how was p adjusted? So can p values of 0.05 be interpreted as “significant” or only those below 0.017?

Discussion

  1. Why should “various marine resources promote the process of awareness and acceptance of the participant, and increased mood, cognitive flexibility, and efficiency” ?

Response:

  • We appreciate your attentive comments. We have vaguely described the content, and have corrected it as follows.
  • In line 408-426: “There were significant time × group interactions on resting pain, movement pain, tactile sense, left grip strength, and physical health and environmental domains of QoL. This difference is presumed to be due to the environmental characteristics of the urban region and marine regions environmental characteristics. The appreciation of natural scenery can promote cortisol reduction and serotonin secretion in the blood, improve mental and physical stability and emotional control, and allow the mind to relax [61]. These physical, psychological, and physiological changes could relieve musculoskeletal pain. Compared to the urban region, the subjects who performed the intervention in the marine region were exposed to many visual stimuli such as marine scenery, auditory stimuli such as the sound of the sea, and tactile stimuli such as sea sand and sea. The sea sand provides an unstable ground to the subject due to its mechanical properties, which require more physical muscle activity to maintain posture and perform movements [19]. Also, in the body in contact with marine resources, sensory improvement can be promoted through more mechanical stimulation and proprioceptive sensory stimulation. In addition, natural ocean sounds can provide relaxation and improvement of mood to subjects as white noise [17]. Stimulation of these various sensory and motor nerves can further promote nerve activation through spatial summation and, together with it, can more effectively improve individual sensory perception. Similar to our results, Choi et al. reported that environmental factors were effective on subjects' upper extremity muscle activity [62].”

>>> In my opinion, this is a very good addition to the Discussion.

  1. Try to offer some suggestion already in the Discussion. There you could also discuss advantages and disadvantages of the chosen locations. E.g., needing more time and spending time away from home, for beach side interventions. Also costs. And maybe also stress more that city location interventions had positive effects.

Response:

  • We thank you for pointing out something we haven't considered. First of all, in this study, both the CG group and the MTG were spent at the hotel and resort during the study period, and the transfer time and intervention time were the same. In addition, all factors other than the environment were adjusted equally between the two groups.
  • We compared the results between the two groups and additionally described the effects of environmental factors on physical, mental, and QoL in the Discussion section.
  • We provided all the expenses for hotel lodging, meals, and hiring a caregiver for all subjects. There is a limitation in clearly identifying the contents pointed out by the reviewer, such as economic factors and the time to move to the arbitration site, only with the results of this study.
  • In our study, meditation and exercise in the city were effective in improving pain and muscle strength, and applying the same intervention in the marine region showed more positive effects in several variables. However, these are all free situations, and it's unclear whether this would have the same effect in a paid context.
  • We have described this issue in the Limitations section. If there is anything we need to correct, please let us know.

>>> Thank you for your detailed explanation. I would mention some concrete implications in your manuscript. E.g., that you recommend the marine therapy. Maybe you can recommend the marine therapy alongside the current recommended therapy for pain. Additionally you could shortly discuss costs for people who want to undergo a marine therapy and then also recommend the “city” therapy instead, because this intervention helped as well.

  1. If city location interventions had so little positive effects, maybe the intervention could be modified and e.g., mediation could be removed? Maybe a sports program and more exercises can be suggested?

Response:

  • The reviewer asked a fascinating question. This is something we are also curious about. First of all, in our study results, CG participated in the study in the urban region and significantly improved resting pain, pinch strength, and depression. Subjects also finished the study and thanked our team for improving their pain. I think this means that meditation and exercise in an urban environment are helpful enough for caregivers.
  • However, from the results of this study, it is not possible to know what effects a sports program or other exercise intervention can have on the physical and psychological health of caregivers in an urban environment. Also, we do not know what the results will be in the situation where the intervention used in this study is performed in an urban forest.
  • We think that we have presented the effects of environmental factors. However, your question is fascinating and will be addressed in future studies. Once again, we thank you very much for your kind question.

>>> Thank you for your detailed comments; I suggests hinting (some of) those questions already in the Discussion. Maybe how could a “low budget” environmental intervention look like; could paintings/sounds/walking on sand be established in urban regions as well. Give an outlook on those possibilities and directions for future studies.

Author Response

Dear Reviewer,

I would like to thank you for providing the opportunity to revise and resubmit the attached manuscript entitled “The Effects of Meditation with Stabilization Exercise in Marine Region on Pain, Tactile sense, Muscle Characteristics and Strength, Balance, Quality of Life, and Depression in Female Family Caregivers of People with Severe Physical Disabilities: A Randomized Single-Blinded Controlled Pilot Study” for publication in IJERPH.

We deeply appreciate the editorial comments and reviewers’ helpful comments on our manuscript which we ignored. We agreed with the points addressed by the Reviewers. We provide our responses to the Reviewers’ comments. Please review the attached files.

Reviewer 2 Report

I read a very different and high quality manuscript. I would like to thank the authors of the manuscript for dealing with my comments in a very effective way. 

Author Response

All authors would like to thank you very much for your good comments. We did our best in the 2nd revision to improve the quality of the paper. We hope that the changes we make will be more satisfying to you.
Again, thank you very much.

Reviewer 5 Report

I agree with the changes made, but I support my previous opinion that the article level is lower than the journal impact factor.

The changes made increased the manuscript level, but the final decision of acceptance belongs to the Editor.

Author Response

First of all, we sincerely apologize for not meeting your standards.
We have been working hard to improve the quality of the paper, and we hope that you will be satisfied with our revisions.